# Dual Mobility Cups as the Routine Choice in Total Hip Arthroplasty

**DOI:** 10.3390/medicina58040528

**Published:** 2022-04-09

**Authors:** Ignacio Aguado-Maestro, Inés de Blas-Sanz, Ana Elena Sanz-Peñas, Silvia Virginia Campesino-Nieto, Jesús Diez-Rodríguez, Sergio Valle-López, Alberto Espinel-Riol, Diego Fernández-Díez, Manuel García-Alonso

**Affiliations:** 1Department of Traumatology and Orthopaedic Surgery, Río Hortega University Hospital, 47012 Valladolid, Spain; ideblassanzd@saludcastillayleon.es (I.d.B.-S.); asanzpe@saludcastillayleon.es (A.E.S.-P.); scampesinon@saludcastillayleon.es (S.V.C.-N.); jdiezrod@saludcastillayleon.es (J.D.-R.); svallel@saludcastillayleon.es (S.V.-L.); aespinelriole@saludcastillayleon.es (A.E.-R.); mgarciaal@saludcastillayleon.es (M.G.-A.); 2Department of Traumatology and Orthopaedic Surgery, Infanta Sofía University Hospital, 28703 Madrid, Spain; dfernandezdiez@gmail.com

**Keywords:** hip, arthroplasty, replacement, dual mobility

## Abstract

*Background and Objectives:* Total hip arthroplasty (THA) is considered the most successful surgical procedure in orthopedics. However, dislocation remains the main indication for surgical revision. New designs of dual mobility cups (DMC) have lowered the classical complications and have extended the indications of DMC in elective surgeries. Our aim is to assess the trend of DMC indications in THA as well as the incidence of their dislocation. *Materials and Methods:* We retrospectively reviewed all patients undergoing THA with DMC during the years 2015 and 2021. The original indication for DMC included patients sustaining neck of femur fractures (NOF#) and associated risk factors for dislocations. Five years later, DMC was considered our standard of care in total hip arthroplasty. The approach (anterolateral or posterolateral) was chosen by the surgeon according to his/her preferences, as was the implant. Data collected included patients’ demographics, diagnosis, admission time, surgical approach, cup models, and inclination and complications. Patients sustaining a hip dislocation were prospectively reviewed and assessed for treatment received, new dislocations, and need for surgical revision. Two groups were created for the analysis according to the presence or absence of dislocation during follow-up. *Results:* In the analysis, 531 arthroplasties were included (mean age 72.2 years) with a mean follow-up of 2.86 years. The trend of indications for DMC increased from 16% of THA in 2015 to 78% of THA in 2021. We found a total of 8 dislocations (1.5%), none of them associated with elective surgery. Closed reduction was unsatisfactory in four cases (50%). There was one case of intraprosthetic dislocation. Dislocations were associated to smaller heads (22 mm) (1.5% vs. 25%, *p* = 0.008) and cups (51.2 mm vs. 48.7 mm, *p* = 0.038) and posterior approach (62.5% vs. 37.5%, *p* = 0.011). *Conclusion:* Dual mobility cups are a great option to reduce the risk of dislocation after a THA both in the neck of femur fractures and elective cases. The use of an anterolateral approach in THA after a neck or femur fracture might considerably decrease the risk of dislocation.

## 1. Introduction

Total hip arthroplasty (THA) is the most successful surgical procedure in our specialty nowadays, as it improves the patients’ quality of life due to the reduction of pain and enhancement of function [1,2]. In the meantime, more than 450,000 THA are performed in the United States every year [3]; whereas in Spain, nearly 46,000 THA were implanted in the year 2017 [4].

Due to the increase in life expectancy, in the year 2030, it is anticipated an increase as high as 174% in the demand for THA in the United States when compared with data from the year 2005 [2], reaching 572,000 hip replacements every year [5].

Dislocation is the most common indication for THA revision during the first year [1,2], with an incidence up to 4% after a primary THA and 25% after a revision surgery [5]. We can divide the risk factors into two groups: those related to the patient and those related to the surgery itself [5,6]. The former include neuromuscular disease, obesity, cognitive impairment [2,5], age above 75 years, previous hip surgery [1], spinal fusion, THA after a femur fracture or vascular necrosis, and rheumatoid arthritis [5]; the latter includes approach (posterior approach is related to a higher incidence of dislocation), malposition of components, inadequate tension of the soft tissues and smaller femoral head implants [5].

Over time, different strategies and techniques have been developed to reduce the risk of instability after THA, such as posterior soft tissue repair in the posterior approach [7]. Moreover, few design modifications of the implants can contribute to avoiding dislocations: higher diameter femoral heads [8] and constrained cups [5]. However, handling instability might be a controversial issue, especially in patients in which the cause is not clear and in those presenting with abductor deficiency [9].

The dual mobility concept was introduced by Gilles Bousquet and Andrè Rambert in France in 1974 to prevent arthroplasties from being dislocated [10]. Dual mobility cups include an additional bearing interposing a mobile polyethylene component (liner) between the femoral prosthetic head and the acetabular shell. It is the perfect combination of Charnley’s principle of low friction and McKee-Farrar concept of increasing stability with a higher diameter femoral head [2,11]. In this type of prosthesis, we can find two articulations: a smaller one between the prosthetic head and the polyethylene, and a bigger joint between the polyethylene head and the acetabular cup. Most of the hip movement will be performed by the smaller joint, while the bigger joint participates in cases with a broader range of motion in which the stem neck gets in contact with the polyethylene head [3,12].

The first designs of dual mobility bearings showed wearing and intraprosthetic dislocation as the main complications, thus their application was indicated in very few cases [3,13]. Wearing might happen in three different zones: smaller and bigger joints, already described, and the surface between the stem neck and the polyethylene head. The majority of the concerns regarding the wearing and survivorship of dual mobility THA involved the latter [14], but in 2011, Vielpeau et al. concluded that wearing is comparable to the observed in standard THA [15]. On the other hand, intraprosthetic dislocation is defined as the separation of the prosthetic femoral head from the polyethylene that holds it, most of the time, secondary to excessive wearing of the polyethylene. Other publications have addressed the cause for this dislocation as: cup loosening, liner blockage by extrinsic factors, or absence of arthrofibrosis [16]. As new designs of dual mobility cups have emerged, several publications have demonstrated that these complications belong to the past and should not determine a contraindication for a dual mobility cup THA [3,15,16].

The aim of our study is to assess the trend of dual mobility cups indications in THA in our department as well as the incidence of their dislocation.

## 2. Materials and Methods

We performed a retrospective analysis of all consecutive patients who underwent total hip replacement with dual mobility cups in our Orthopedic Surgery Department between 2015 and the first quarter of 2021. In our department, we started using dual mobility cups in 2015, indicated at first in patients sustaining intracapsular fractures and presenting factors associated with dislocation such as frailty, limb weakness or paresis, Parkinson’s disease, or other neurologic disorders. The indication for dual mobility cups was extended to revision surgeries and all intracapsular fractures in 2016. In 2017, we used dual mobility cups in primary total hip replacements in cases of secondary osteoarthritis due to hip dysplasia or Perthes disease. Indications were progressively broadened and three years later, in 2020, dual mobility cups were considered our standard of care in total hip arthroplasty [10,17]. All surgeries were performed under spinal anesthesia with hyperbaric bupivacaine unless contraindicated or previous spine fusion surgery. The approach (anterolateral or posterolateral) was chosen by the surgeon according to his/her preferences, as was the implant (Avantage^®^Dual Mobility Cup System, Zimmer-Biomet, Warsaw, Indiana, USA; Apogée^®^Dual Mobility Cup, Biotechni, La Ciotat, France; G7^®^Acetabular System, Zimmer-Biomet, Warsaw, IN, USA). In all the cases, a cementless technique was chosen, both for the cup and the stem. One day after the surgery, a blood test and an X-ray was taken. If hemoglobin levels were above 8.5 g% and the X-ray was satisfactory, patients were allowed to seat on a chair. Ambulation was usually started 48 h after surgery with the aid of two walking crutches or a walking frame according to patients’ capabilities.

Data collected included patients’ demographics, main diagnosis, laterality, admission time, surgical approach, cup models, cup angulation in the anteroposterior (AP) radiological view, surgical complications (during surgery and in the post-op), and other complications. A minimum follow-up of 12 months was set. Inclusion criteria included all patients in which a dual mobility cup was implanted during surgery. Patients in which a single mobility cup was used or with a follow-up of less than 12 months were excluded from the analysis. 

As the main objective of our study was to assess the incidence of dislocation of dual mobility cups, patients sustaining a hip dislocation were prospectively reviewed and assessed for treatment received, new dislocations, and need for surgical revision. Two groups were created for the analysis according to the presence or absence of dislocation during follow-up.

### Statistical Analysis

Data was collected in a Microsoft^®^Excel spreadsheet (Microsoft^®^Excel for Mac v. 16, Microsoft, Redmond, WA, USA) and statistical analysis was performed with SPSS (IBM^®^ SPSS^®^ Statistics for Mac v.25, Armonk, NY, USA). Parametric and non-parametric tests were used as required. Significance was considered for a *p*-value below 0.05.

## 3. Results

A total of 531 arthroplasties were included in the analysis. The mean age was of 72.2 years [range: 19–93; standard deviation (SD): 11.7], 198 (37.3%) male and 333 (62.7%) female. The mean follow-up was 2.86 years [range: 0.5–6.94 years; SD: 1.69]. The distribution of the implanted cups was as follows: Avantage: 79 cases (14.9%), G7: 365 cases (68.8%), and Apogée: 87 cases (16.4%).

The trend in the indications for dual mobility cups is shown in Figure 1 as a comparison between the total hip replacements performed in our department and the cases in which dual mobility cups were implanted. We found a total of 8 dislocations (1.5%). A closed reduction under general anesthesia was attempted in all cases. However, this was unsatisfactory in 4 cases (50%) which needed an open reduction. The open reduction was associated with other surgical techniques such as implant revision, removal, or osteosynthesis in 3 cases. (Table 1).

We then proceeded to compare both groups (those sustaining a dislocation and those who did not) in order to find possible risk factors related to this event. The results are shown in Table 2.

Statistically significant results were observed regarding the primary diagnosis, as all the dislocation events appeared in patients who had presented with an intracapsular neck of femur fracture (NOF#) (*p* = 0.003). Dislocations were more frequent in patients who had been implanted with smaller heads (22 mm) (1.5% vs. 25%, *p* = 0.008) and cups (51.2 mm vs. 48.7 mm; *p* = 0.038) as well as in patients operated through a posterior approach (62.5% vs. 37.5%; *p* = 0.011). The incidence of dislocation with regard to the approach was 4.6% in the posterior approach and 0.7% in the anterolateral approach. The mean time until the first dislocation was of 67.6 days [1–376 days; SD: 126.92]. Half of the patients underwent a satisfactory closed reduction under general anesthesia, whereas the other half needed an open surgical reduction. In three cases, the dislocation was presented during the admission time after hip replacement surgery. In three cases, dislocation was recurrent. This was more frequent in those cases with early presentation (mean time until the first dislocation in recurrent instability: 17.33 days). There was one case of intraprosthetic dislocation, which appeared after minor trauma (simple fall) 376 days after the index surgery. A closed reduction was attempted, but while the ceramic head could be introduced inside the cup, the polyethylene remained outside of it (Figure 2 and Figure 3). 

Hospital admission time was longer in patients who presented dislocation (7.1 vs. 11.7 days; *p* < 0.001) as could be expected due to the early occurrence in 3 cases, during admission time of the index hip replacement surgery.

## 4. Discussion

Our results are comparable to those described in the bibliography with a dislocation rate of 0% in primary total hip arthroplasty and 3.1% after a NOF#. In the systematic review and meta-analysis conducted by Cha et al., the overall dislocation rate of dual mobility cups after a NOF# was 4% [18]. This result is slightly higher than the 2.7% reported by Cnudde et al. with data from the Swedish Registry regarding patients with a neurological disease who sustained a NOF# [19]. Concerns regarding the generalized use of dual mobility cups have arisen especially in relation to intraprosthetic dislocation [3,20] and the possibility of excessive wearing [5]. Contemporary dual mobility cups including a better design and more important, stabilized and highly cross-linked polyethylene, have contributed to the spread of the indications including primary total hip arthroplasties, with excellent results in the cohort study reported by Epinette with a dislocation rate of 0% favoring dual mobility cups [21]. Other authors such as Vigdorchik et al. reproduced these results in the United States [22]. The trend has been noted as well in our very own department, in which we started using dual mobility cups in patients with an increased risk of dislocation [23,24] such as cerebral palsy, hip dysplasia, Perthes’ disease, and NOF# extending the indications as long as good results were reported and also observed and analyzed in our own series of patients. Nowadays, dual mobility is the choice of most surgeons in their primary THA, as new cup designs offer good press-fit with primary fixation and the versatility of holes for screws when needed [1]. Moreover, its use has been described as cost-effective [25,26].

With regards to the approach, a statistically significant result was observed, being the posterior approach a risk factor for dislocation after THA (4.6 vs. 0.7%) as was described by Matharu et al. with data from the National Joint Registry from England, Wales, Northern Island, and the Isle of Man [27,28] and was confirmed to be the most important risk factor for dislocation in the study of Cnudde et al., which involved dual mobility cups [19]. 

Bigger heads (and hence, bigger cups) significantly reduce the risk for dislocation in elective THA. This is a commonly accepted rule that has been described in the medical literature before [29]. However, there is still controversy regarding the relationship between the size of the head and the risk of dislocation when THA is indicated after a NOF# [30]. In our series, when adjusting for indication, we did observe a statistically significant higher risk of dislocation when the head diameter was 22 mm instead of 28 mm (*p* = 0.034). 

Modular cups were chosen in 68.8% of patients and were responsible for 7 (87.5%) of the observed dislocations (*p* = 0.445). Although our results are not significant, there is a trend in the European orthopedic community led by Aslanian and Tigani which suggest that modular cups are related to an increased risk of dislocation due to the reduction of the jumping distance secondary to the significant increase of the global thickness of the metal cup [31,32,33,34]. 

We had one case of intraprosthetic dislocation. We cannot confirm whether it was produced during the traumatic event that produced the dislocation or during the closed reduction maneuvers, as the CT scan was only performed after the reduction attempt (Figure 2). Theories regarding this event include polyethylene wear as the leading cause, followed by aseptic loosening or blocked articulation between the liner and the metal cup due to arthrofibrosis or heterotopic ossifications [3,20]. Additionally, modular cups have been related to intraprosthetic dislocations due to the possibility of corrosion between the titanium cup and the chrome-cobalt metal insert [31]. Early intraprosthetic dislocations have been considered by these authors to be caused by a mechanical failure of the retentive rim. As there were only 376 days between the dislocation and the index surgery, with normal X-rays during follow up, we do believe that it might have been an iatrogenic intraprosthetic dislocation, in which the liner might have been gripped by the edge of the cup during the reduction maneuvers of the primary dislocation [3]. This event might have been influenced by the design of the retentive polyethylene [32].

Our research limitations involve the low number of dislocation events observed (*n* = 8) which may not be enough to support all the research findings. 

## 5. Conclusions

Dual mobility cups are a great option to reduce the risk of dislocation after a THA both in the neck of femur fractures and elective cases. The use of an anterolateral approach in THA after a neck or femur fracture might considerably decrease the risk of dislocation.

## Figures and Tables

**Figure 1 medicina-58-00528-f001:**
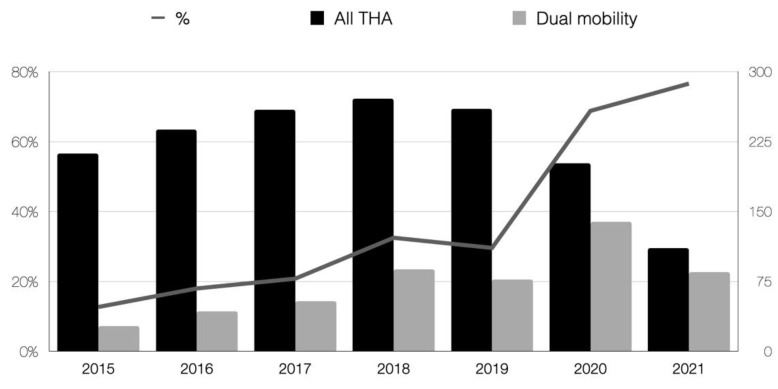
Trend in the indications for total hip replacements with dual mobility cups in our department as a comparison with the total amount of total hip replacements performed. The line shows the proportion.

**Figure 2 medicina-58-00528-f002:**
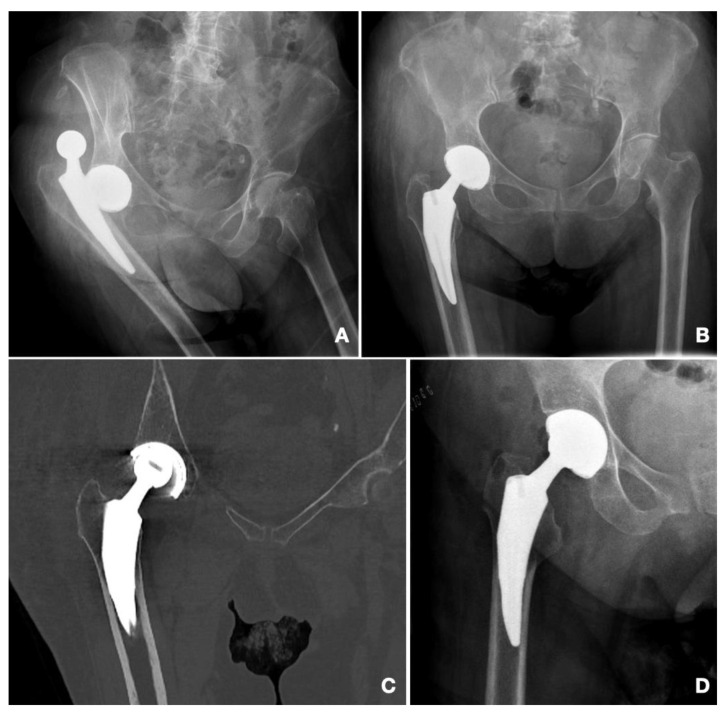
Case number 7. Intraprosthetic dislocation. (**A**) Emergency department X-ray. (**B**) Control X-ray after closed reduction maneuvers. (**C**) CT Scan after closed reduction maneuvers. Note the displacement of the head, eccentric within the acetabular cup. (**D**) Post-op control after revision surgery.

**Figure 3 medicina-58-00528-f003:**
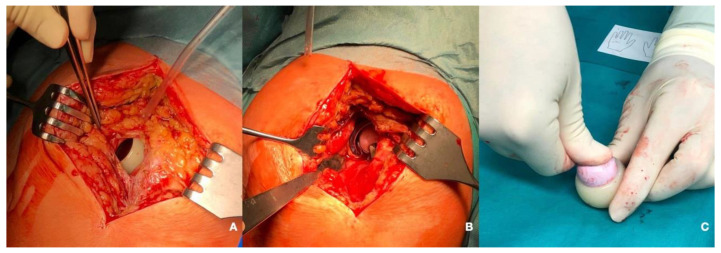
Intraoperative photographs show the dislocated liner (**A**) and the reduced head in the acetabular cup (**B**). An attempt to introduce manually the head in the liner to check for an abnormal weakness or defect was performed but was not possible (**C**).

**Table 1 medicina-58-00528-t001:** Summary of all patients sustaining a dislocation.

ID	Cup	Time after Index Surgery	Mechanism	Reduction	Redislocation	Remarks	Need for Revision	Functional Status (Parker)
1	G7	5	Atraumatic	Closed	-		No	Walks with one simple aid
2	G7	45	Atraumatic	Closed	26 days after	Death < 1 year	No	Walks with walking frame
3	G7	1	Atraumatic	Open	4 days after	Death < 1 year	1st episode: Stem and Cup revision 2nd episode: Girdlestone	No walking
4	Apogee	13	Traumatic	Open				No walking
5	G7	6	Atraumatic	Closed	1 day after 9 days after		2nd episode: Cup revision 3rd episode: Cup revision	No walking
6	G7	72	Atraumatic	Closed				Walks with one simple aid
7	G7	376	Traumatic	Open		Intraprosthetic dislocation	Revision of dual mobility head	Walks with walking frame
8	G7	23	Atraumatic	Open		Dislocation associated with Vancouver B2 fracture	Revision of stem	Walks with one simple aid

**Table 2 medicina-58-00528-t002:** Risk factors associated with a dislocation event. Group A: patients who did not sustain a dislocation). Group B: patients who presented a dislocation during follow-up.

	Group A (Non Dislocated) *n* = 523	Group B (Dislocated) *n* = 8	*p* Value
Age (years)	72.0 (SD: 11.7)	78.49 (SD: 5.96)	0.122
Sex (%)	Male: 197 (37.6%) Female: 326 (62.4%)	Male: 2 (0.25%) Female: 6 (0.75%)	0.716
Side	Left: 261 (49.9%) Right: 262 (51.1%)	Left: 4 (50%) Right: 4 (50%)	1
Indication	NOF#: 247 (47.2%) HOA: 276 (52.8%)	NOF#: 8 (100%) HOA: 0	0.003
Head size (mm)	28 mm: 515 (98.5%) 22 mm: 8 (1.5%)	28 mm: 6 (75%) 22 mm: 2 (25%)	0.008
Cup size (mm)	51.2 (SD: 3.34)	48.7 (SD: 2.81)	0.038
Cup inclination (°)	44.8 (SD: 7.86)	41.9 (SD: 6.83)	0.299
Surgical time (min)	73.8 (SD: 23.72)	75 (SD: 13.09)	0.890
Approach	WJ: 421 (80.5%) PL: 102 (19.5%)	WJ: 3 (37.5%) PL: 5 (62.5%)	0.011
Length of stay (days)	7.1 (SD: 3.43)	11.7 (SD: 7.74)	<0.0001
Mortality (1 year)	25 (4.7%)	2 (25%)	0.06

NOF#: Neck of Femur Fracture (intracapsular). HOA (Hip Osteoarthritis, primary or secondary to Perthes Disease, Dysplasia, Avascular Necrosis…). WJ: Watson Jones anterolateral modified approach. PL: Posterolateral Moore approach.

## Data Availability

The data presented in this study are available on request from the corresponding author.

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
