# Peer review of "Dual Mobility Cups as the Routine Choice in Total Hip Arthroplasty"

_medicina, 2022, doi:10.3390/medicina58040528_

Round 1

Reviewer 1 Report

Dear Authors

I have reviewed with great interest your article on Dual Mobility.

The form and content of this publication is completely consistent and requires only minor corrections.

On the other hand, the detailed analysis of the cases that presented a dislocation is quite demonstrative and astonishing because all the dislocations and the intraprosthetic dislocation occurred on a Zimmer G7 cup, which for us French surgeons who are used to double mobility is a design "heresy" for the following reasons

1 it is a MODULAR cup intended essentially for the North American market, which does not conceive of the use of a cementless cup WITHOUT additional fixation with screws. The modularity has 2 perverse effects (a reduction of the jumping distance by a significant increase of the global thickness of the metal cup, a sub hemispherical Metal back design which reduces the jumping distance even more. I sincerely invite you to read the very nice article by Nevelos which describes very well this design specificity.

2 The PE insert is made of Higly cross linked PE, which is very efficient in terms of wear, but unsuitable for dual mobility because its intrinsic rigidity produces micro-fractures in the retention device, significantly increasing the risk of intra-prosthetic dislocation.

3 the modularity is a risk factor for corrosion between the Titanium back metal and the Chrome Cobalt metal insert.

4 All recent case reports published in the literature concerning intraprosthetic dislocation are observed on implants of this type (Modular DM such as Zimmer G7 or MDM Stryker).

It is surprising that you did not ask yourself the question of the type of implant that was dislocated, because your analysis is perfectly demonstrative.

I am almost convinced that if you use European implants that respect the philosophy of G. Bousquet in association with a round and polished femoral prosthetic neck, this complication will disappear from your statistics for good.

In conclusion, I invite you to rewrite your discussion by arguing very precisely about the danger of using this type of design, which will be of great service to the orthopaedic community, which sometimes forgets the basic principles and could be seduced by marketing gimmicks.

Tigani D, Prudhon JL, Amendola L, Aslanian T. Letter to the editor on "Early intraprosthetic dislocation in dual-mobility implants: a systematic review". Arthroplast Today. 2017 Dec 13;4(1):132. doi: 10.1016/j.artd.2017.11.004. PMID: 29560409; PMCID: PMC5859209.

Aslanian T. All dual mobility cups are not the same. Int Orthop. 2017 Mar;41(3):573-581. doi: 10.1007/s00264-016-3380-3. Epub 2017 Jan 18. PMID: 28097387.

Nevelos J, Johnson A, Heffernan C, Macintyre J, Markel DC, Mont MA. What factors affect posterior dislocation distance in THA?. Clin Orthop Relat Res. 2013;471(2):519-526. doi:10.1007/s11999-012-2559-1

Heffernan C, Banerjee S, Nevelos J, Macintyre J, Issa K, Markel DC, Mont MA. Does dual-mobility cup geometry affect posterior horizontal dislocation distance? Clin Orthop Relat Res. 2014 May;472(5):1535-44. doi: 10.1007/s11999-014-3469-1. Epub 2014 Jan 24. PMID: 24464508; PMCID: PMC3971207.

Author Response

Thank you very much for your kind review. Without any doubt your comments have enriched our article. Merci beaucoup!

We have addressed the suggested changes in our Discussion.

As we are also European, we started using “European” designs such as the apogee and avantage. However, the difficulty found by certain nurses trying to engage the cup for its insertion made most of the surgeons to switch to the G7.

Thank you very much again, I hope you find our ammendments satisfactory.

Yours sincerely

Reviewer 2 Report

Dear Author(s),

Please present p value as p<0.001 (instead of p<0,0001) regarding the following sentence: "Hospital admission time was longer in patients who presented dislocation (7.1 vs 11.7 182 days; p<0,0001)".

The aim of our study is to assess the trend of dual mobility cups indications in THA in their department and the incidence of their dislocation. Authors demonstrated well the primary aim.

Weak evidence due to the nature of the study design - The author(s) found a total of 8 dislocations (1.5%), and conducted the association analysis regarding a dislocation event (Table 2). However, N=8 seems not enough sample size to support these research findings; dislocations were associated to smaller 26 heads (22 mm) (1.5% vs 25%, p=0,008) and cups (51.2 mm vs 48.7 mm, p=0.038) and posterior approach (62.5% vs 37.5%, p=0,011). Needs to be stated clearly about this limitation.

Author Response

Dear Reviewer,

We would like to kindly thank you for your comments. We have addressed your all your proposed changes and have included a "limitations" statement in our discussion. 

Yours sincerely.

Round 2

Reviewer 1 Report

Thanks to the authors for the corrections suggested in the previous report. 

article accepted